# Expert-annotated datasets for deep-learning based classification methods for morphologic Leukaemia diagnostics

**Christian Matek** [1,2]                    CHRISTIAN.MATEK@UK-ERLANGEN.DE

**Carsten Marr**[2]                    CARSTEN.MARR@HELMHOLTZ-MUNICH.DE

[1] *Institute of Pathology, University Hospital Erlangen, Friedrich-Alexander-Universität Erlangen-Nürnberg (FAU), Erlangen, Germany*

[2] *Institute of AI for Health, Helmholtz Munich, Neuherberg, Germany.*

## Abstract

In recent years, image analysis and classification methods have become increasingly popular for segmentation and classification tasks in pathology imaging, specifically for the evaluation of tissues sections and single cells.

Hematological cytomorphology, with single cell-classification at its base and lacking some of the complexities of tissue architecture, has been at the forefront of this development. Recent additions of large-scale datasets to the public domain open up new possibilities for developing and benchmarking algorithms for morphologic diagnostics, prognostication and subgrouping.

**Keywords:** Microscopy imaging, deep learning, single-cell data, multiple-instance learning.

## 1. Introduction

Over the past 150 years, morphologic examination of blood and bone marrow cells using light microscopy has formed the basis of leukemia diagnostics. Presently, high-resolution microscopy of peripheral blood and bone marrow samples still represents an important step in the diagnostic workup of patients suffering from hematologic malignancies (Means , ed.). Used in conjunction with more recently developed immunologic and genetic tests in modern routine diagnostics, cytomorphology still stands out for its accessibilty, comparatively low cost and wide availability.

A notable problem of morphologic evaluation in current routine practice is its heavy use of trained human staff. Firstly, readling cytology specimens without preselection of diagnostically relevant areas can be a tedious and time consuming task. Secondly, many diagnostic settings worldwide suffer from a lack of trained examiners experienced in cytomorphology. Finally, the intrinsically subjective nature of morphologic classificaiton can lead to significant inter- and intra-rater variabiliy of results (Fuentes-Arderiu and Dot-Bach; Font et al.). Hence implementing robust and widely applicable diagnostic decision support systems for cytomorphologic examination corresponds to a relevant clinical need. Recent advances in deep learning-based image classification offer promising methods for developing such tools. As for many data-driven methods in medical diagnostics, progress in algorithm development can be limited by the lack of publicly available high-quality datasets. Here we present a set of recently published curated reference datasets containing microscopic images of blood and bone marrow cells, as well as a set of algorithms developed based on these datasets which answer diagnostically and therapeutically relevant questions.

## 2. Datasets

Recently, a number of datasets on hematological cytomorphology have been published for both peripheral blood and bone marrow. Specifically, datasets related to work from our group were anonymized in concordance with the relevant ethics votes, and deposited in The Cancer Imaging Archive (TCIA) (Clark et al.). Relevant dataset parameters are summarized in Tab. 4, and sample images from two datasets are shown in Fig. 1. Refs. (Matek et al., c) and (Hehr et al., b) cover single-leukocyte images from peripheral blood, while ref. (Matek et al., a) contains microscopic images of bone marrow smears, which represents the diagnostic gold standard. All three image databases are curated, and ref. (Hehr et al., b) additionally contains clinical patient information. While the peripheral blood datasets focus on acute myeloid leukemia (AML), ref. (Matek et al., a) contains bone marrow samples from a wide range of hematologic diseases detailed in ref. (Matek et al., b). While expert-annotated datasets have been published before for peripheral blood (Acevedo et al.) and - more rarely - for bone marrow samples (Choi et al.), these to the authors' knowledge represent the largest publicly available databases today, providing a benchmark and resource for future algorithm development.

## 3. Classification approaches

Expert annotations accompanying the datasets described in this short paper allow training and evaluating algorithms for both single-leukocyte classification and patient-level subtyping. Specifically, we used the ResNeXt architecture in order to train a classifier for leukocytes in AML which attains human-level performance for blast cell recognition (Matek et al., d). For differention of leukocytes relevant in the bone marrow, a classifier can be trained that shows similar performance (Matek et al., b). Finally, a classifier can be trained to predict the genetic subtypes of AML from a bag of leukocyte images using attention-based multiple instance learning (Hehr et al., a). Predictions are amenable to explainability methods, both for single and multiple instance classifiers (Matek et al., b; Sadafi et al., a). All single-leukocyte classifiers obtain a reduced accuracy for minority classes, reflecting the importance of class imbalance. Variable imbalance between different domains can be accounted for using cross-domain feature extraction (Salehi et al.) or dedicated domain-generalization algorithms (Umer et al.). Additionally, trained algorithms can be sensitive towards biased patient age and biological sex distributions (Sadafi et al., b).

## 4. Conclusions and outlook

Comprehensive datasets of malignant and non-malignant leukocyte classes allow harnessing the power of current deep learning-based image classification algorithms to train classifiers that achieve a high diagnostic accuracy in single-cell classification of many relevant cell classes in both peripheral blood and bone marrow. For some specific classification tasks, human-level performance is achieved. Major challenges in routine implementation of these methods are a significant class imbalance between individual cell types, an effect due to the biological distribution of relevant leukocyte types, as well as domain-specific staining and cell-type distribution differences. These challenges can be met by utilizing dedicated

domain-generalization methods, as well as increasing the size and diversity of datasets used as training data.

| Dataset | Individuals | Leukocytes | Resolution | Reference |
|---|---|---|---|---|
| AML-Cytomorphology_LMU | 200 | 18,365 | 0.07072 $\mu m/px$ | (Matek et al., c) |
| AML-Cytomorphology_MLL_Helmholtz | 189 | 81,214 | 0.1725 $\mu m/px$ | (Hehr et al., b) |
| Bone-Marrow-Cytomorphology_MHF | 945 | 170,000 | 0.086 $\mu m/px$ | (Matek et al., a) |

Table 1: Overview of three reference datasets for leukemia cytomorphology deposited in The Cancer Imaging Archive (TCIA).

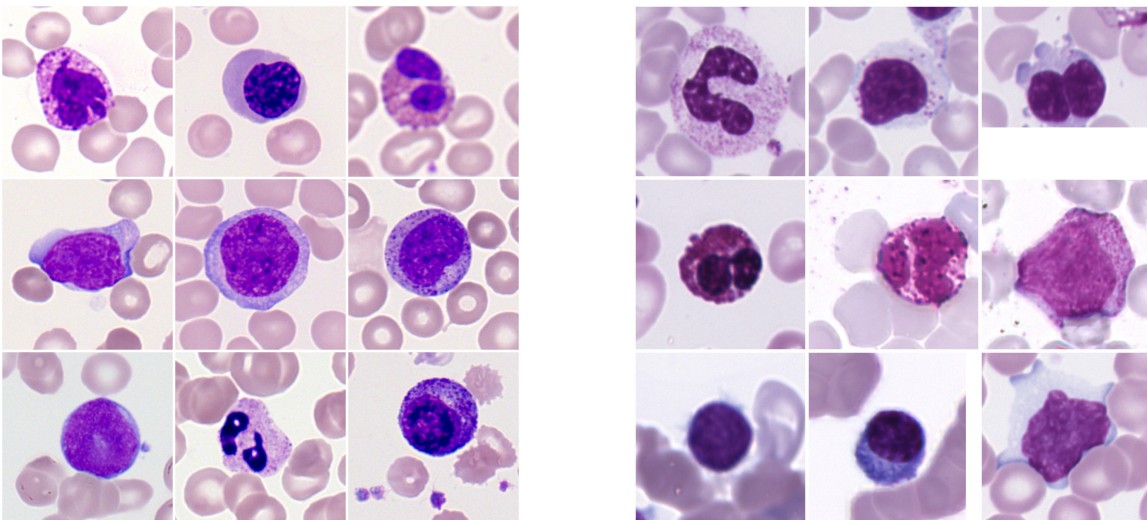

Figure 1: Sample images of leukocytes from peripheral blood smears (left, (Matek et al., c)) and bone marrow (right, (Matek et al., a))

## Acknowledgments

Both authors acknowledge funding from the European Research Council (ERC), via grant agreement no. 866411 to Carsten Marr.

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
