# OpenReview forum: "Expert-annotated datasets for deep-learning based classification methods for morphologic Leukaemia diagnostics"
_MIDL.io/2024/Short_Papers — MIDL 2024 Short Papers_

### Official Review · Reviewer_3RR5 · 2024-04-24

**Confidence:** 4
**Final Rating:** 4

**Review:**

This paper introduces a set of three datasets of images for leukaemia cytomorphology analysis, which has been made publicly available via the TCIA platform.

#### PROS
* The public release of annotated data is relevant to the field
* The paper reports a set of classification approaches that have been developed by the authors of this short paper, with results achieved on the proposed datasets (e.g., single leukocyte classification, patient-level subtyping).

#### CONS
* No results are directly presented in this short paper, mostly references to previous work from the same authors; in the spirit of MIDL short papers, this is acceptable in my opinion, but covering some more specific aspects of classification results would have improved the quality of this paper
* No detailed description of data collection and annotations is described in this short paper (also partly understandable given the 3-page limit)

---

### Decision · Program_Chairs · 2024-04-26

Accept